# Silver Nanoparticle–PEDOT:PSS Composites as Water-Processable Anodes: Correlation between the Synthetic Parameters and the Optical/Morphological Properties

**DOI:** 10.3390/polym15183675

**Published:** 2023-09-06

**Authors:** Stefania Zappia, Marina Alloisio, Julio Cesar Valdivia, Eduardo Arias, Ivana Moggio, Guido Scavia, Silvia Destri

**Affiliations:** 1Istituto di Scienze e Tecnologie Chimiche “Giulio Natta” (SCITEC), Consiglio Nazionale delle Ricerche (CNR), Via Alfonso Corti 12, 20133 Milano, Italy; silvia.destri@scitec.cnr.it; 2Dipartimento di Chimica e Chimica Industriale (DCCI), Università di Genova, Via Dodecaneso 31, 16146 Genova, Italy; 3Centro de Investigación en Química Aplicada (CIQA), Boulevard Enrique Reyna 140, Saltillo 25294, Mexicoeduardo.arias@ciqa.edu.mx (E.A.)

**Keywords:** PEDOT:PSS, silver nanoparticles, anode, organic photovoltaics, nanocomposites

## Abstract

The morphological, spectroscopic and rheological properties of silver nanoparticles (AgNPs) synthesized in situ within commercial PEDOT:PSS formulations, labeled PP@NPs, were systematically investigated by varying different synthetic parameters (NaBH_4_/AgNO_3_ molar ratio, PEDOT:PSS formulation and silver and PEDOT:PSS concentration in the reaction medium), revealing that only the reagent ratio affected the properties of the resulting nanoparticles. Combining the results obtained from the field-emission scanning electron microscopy analysis and UV-Vis characterization, it could be assumed that PP@NPs’ stabilization occurs by means of PSS chains, preferably outside of the PEDOT:PSS domains with low silver content. Conversely, with high silver content, the particles also formed in PEDOT-rich domains with the consequent perturbation of the polaron absorption features of the conjugated polymer. Atomic force microscopy was used to characterize the films deposited on glass from the particle-containing PEDOT:PSS suspensions. The film with an optimized morphology, obtained from the suspension sample characterized by the lowest silver and NaBH_4_ content, was used to fabricate a very initial prototype of a water-processable anode in a solar cell prepared with an active layer constituted by the benchmark blend poly(3-hexylthiophene) and [6,6]-Phenyl C61 butyric acid methyl ester (PC_60_BM) and a low-temperature, not-evaporated cathode (Field’s metal).

## 1. Introduction

In the last decade, polymer-based solar cells (PSCs, hereafter) have attracted enormous interest as next-generation green energy sources because of their promising potential due to the cheap cost of manufacturing, flexibility, roll-to-roll and large area processing. From this point of view, electrodes represent a bottleneck for the printing and high-speed processing of PSCs [1,2,3]. 

Poly(3,4-ethylenedioxythiophene) polystyrene sulfonate (PEDOT:PSS) is currently the most successful water-processable conductive polymer with the highest reported conductivity [2,4], which makes it essential in the fabrication of photovoltaics, optoelectronics, bioelectronics, printed electronics and so on [5]. PEDOT is a polythiophene derivative with high stability and conductivity that could be obtained from the oxidative polymerization of the corresponding 3,4-ethylenedioxythiophene (EDOT) monomer. Unfortunately, the obtained PEDOT displayed poor solubility in polar solvents. To overcome this problem, Bayer’s scientists optimized the synthetic procedure by oxidizing EDOT monomers in the presence of the water-soluble sodium salt of polystyrene sulfonate (PSS) [6]. In the resulting stable aqueous dispersion, PEDOT was stabilized with PSS as a counter anion, which also acted as a dopant for the PEDOT backbone [5,7].

As is known, the PEDOT molecular weight is much lower than that of PSS, and several PEDOT oligomeric chains interact with a single PSS chain [8]. In addition, the stability of the aqueous suspension is due to the shielding of the hydrophobic PEDOT by PSS, leading to the formation of grain structures via hydrophilic, PSS-rich shell and PEDOT chains arranged in the core. The grains are linked together with hydrogen bonding between the PSS shell according to a model proposed by Dupont et al. [9]; thus, the content of PSS is strictly connected to the conductivity as it influences the charge-carrier mobility [10].

The PEDOT:PSS film has the advantage of low cost, high transmittance, stability after film formation, and adjustable conductivity [11,12,13]. For these reasons, it has been long-since used as both a hole-transporting layer (injection for organic light-emitting diodes and extraction for PSCs) and an anode in different devices [14,15,16,17]. Highly conductive H_2_SO_4_-treated PEDOT:PSS films replaced the indium tin oxide (ITO) layer as the transparent anode in PSCs, achieving a power conversion efficiency (PCE) of 3.56%, which was comparable to that of the control devices using an ITO anode [17]. Large-area and flexible PSCs need ITO-free anodes due to the limited availability of ITO, as well as its rather high cost, its stiff and brittle nature which generates cracks when the material is subjected to repeat tensile and compressive bending, and finally, its lack of adhesion to organic and polymeric materials.

Besides highly conductive PEDOT:PSS, other materials have been tested for this aim. Among them, silver nanowires (AgNWs) have been considered as a promising alternative due to their excellent electrical conductivity and mechanical flexibility. However, the junction resistance between nanowires and the remaining non-conducting area in the electrode layer may worsen the device performance. A simple method to overcome these drawbacks is the fabrication of a AgNW/PEDOT:PSS nanocomposite obtained by depositing a highly conductive PEDOT:PSS layer on a AgNW-coated substrate followed by annealing [18,19].

Another nanocomposite approach has been applied which involved directly embedding silver nanoparticles (AgNPs) in highly conductive PEDOT:PSS. The resulting device led to a 32% increase in PCE compared to the device without AgNPs [20].

In that research, the AgNPs were prepared separately and then added to PEDOT:PSS, whereas other authors used a different strategy, synthetizing AgNPs directly in PEDOT:PSS via the reduction of AgNO_3_ salt [21] and applying the film of this material as the anode [22]. The electrical and electronic properties of the AgNPs/PEDOT:PSS composite strongly depend on the AgNP size and shape, which in turn are influenced by the AgNO_3_/reducing agent molar ratio and the procedure protocols [22]. Under this context, in the present work, we discussed the leverage of some parameters over AgNP preparation in a PEDOT:PSS matrix, i.e., the AgNO_3_/reducing agent molar ratio, the features of the commercial starting PEDOT:PSS suspensions, and the concentration of the conducting polymer and the silver salt in the reaction medium. In particular, we performed the spectroscopic and morphological characterization of the aqueous colloidal suspensions prepared at different AgNO_3_/reducing agent molar ratios. Then, we performed the morphological and electrical study of the films obtained from the deposition of the dispersions varying these parameters. This empirical study allowed us to achieve a prototypical photovoltaic device characterized by AgNP/PEDOT:PSS composites as the anode and the benchmark blend phenyl-C61-butyric acid methyl ester:poly(3-hexylthiophene) (PC61BM:P3HT) as the active layer. Moreover, Field’s metal was used as the cathode through easy transfer on the active layer in a molten state at 70 °C.

## 2. Materials and Methods

### 2.1. Chemicals

The poly(3,4-ethylenedioxythiophene) polystyrene sulfonate (PEDOT:PSS) aqueous suspensions used in this work were purchased from Sigma-Aldrich (Merck Life Science S.r.l., Milan, Italy), product name Orgacon^TM^ ICP1050 (Agfa, Motzel, Belgium) (product code #739332), and from Heraeus (Heraeus Italia, Monza-Brianza, Italy), product name Clevios^TM^ PH1000, and are referred to hereafter as Orgacon and Clevios, respectively.

The phenyl-C61-butyric acid methyl ester (PC61BM) and poly(3-hexylthiophene) (P3HT) used for the active layers were purchased from Sigma-Aldrich (Merck Life Science S.r.l., Toluca, Mexico) and were used as received without further purifications. ITO slides were purchased from Ossila (Ossila Ltd., Sheffield, UK).

Sodium borohydride (NaBH_4_, Sigma-Aldrich, Merck Life Science S.r.l, Milan, Italy) and silver nitrate (AgNO_3_, Alfa Aesar, Thermo Fisher GmBH, Kandel, Germany) were commercial products used as received.

Aqueous solutions were prepared with ultra-high-purity Milli-Q water distilled twice prior to use.

Freshly prepared “piranha” solution, obtained by mixing concentrated sulfuric acid and cooled hydrogen peroxide (30% *v*/*v*) in the ratio of 2:1 *v/v*, was used to thoroughly clean the glassware before the silver nanoparticles’ synthesis.

### 2.2. Synthesis of Silver Nanoparticles in Presence of PEDOT:PSS

Silver nanoparticles stabilized with PEDOT:PSS (PP@AgNPs) were synthesized “in situ” following different protocols for the wet chemical reduction of AgNO_3_. The first series of samples was synthetized with a method adapted from a previous set-up for the preparation of AgNO_3_ embedded in a polysaccharide matrix [23,24,25]. Briefly, 10 mL of Orgacon was placed in a thoroughly cleaned flask, and 0.1 g of AgNO_3_ was added. The mixture was kept under stirring conditions at room temperature until the complete solubilization of the silver salt. Then, a proper aliquot of a freshly prepared solution of NaBH_4_ in water (1.2 mol/L) was introduced drop by drop and the mixture was (eventually) diluted to 15 mL with water. The procedure was repeated three times to obtain different molar ratios between NaBH_4_ and AgNO_3_. In detail, 5 mL of NaBH_4_ (1.2 mol/L) was added in the PP@AgNPs-1 sample (NaBH_4_/AgNO_3_ ratio = 9.9), 2.5 mL of NaBH_4_ (1.2 mol/L) and 2.5 mL of water were added in the PP@AgNPs-2 sample (NaBH_4_/AgNO_3_ ratio = 4.9), 0.5 mL of NaBH_4_ (1.2 mol/L) and 4.5 mL of water were added in the PP@AgNPs-3 sample (NaBH_4_/AgNO_3_ ratio = 1.0). The mixtures immediately turned from colorless to deep green and were maintained under constant stirring conditions overnight to ensure a full reaction. The end-products were aqueous suspensions with nominal concentrations of ~40 mmol/L in terms of Ag content and were stored at room temperature.

The second series of samples was synthetized according to ref. [26]. Briefly, 10 mL of PEDOT:PSS was placed in a thoroughly cleaned flask and 0.1 g of AgNO_3_ were added. The mixture was kept under stirring conditions at room temperature for 3 h, and then, a freshly prepared aqueous solution of NaBH_4_ (12 mmol/L, 5 mL) was added. After 30 min of stirring, the end-product was an aqueous suspension with a nominal concentration of 40 mmol/L in terms of Ag content and a 0.1 molar ratio between NaBH_4_ and AgNO_3_, which was stored at room temperature. The procedure was repeated twice using both formulations of PEDOT:PSS to obtain samples PP@AgNPs-4 and PP@AgNPs-5, respectively.

The third series of samples was synthetized with an adaptation of the first two methods carried out at a lower concentration of both silver salt and PEDOT:PSS. Briefly, 20 mL of an aqueous solution of AgNO_3_ (5 mmol/L) was placed in a thoroughly cleaned flask and 1 mL of Clevios was added under constant stirring conditions. Then, a proper aliquot of a freshly prepared solution of NaBH_4_ in water (45 mmol/L) was introduced drop by drop and the mixture was diluted to 50 mL with water. The procedure was repeated twice to obtain different molar ratios between NaBH_4_ and AgNO_3_. In detail, 0.25 mL of NaBH_4_ (45 mmol/L) was added in the PP@AgNPs-6 sample (NaBH_4_/AgNO_3_ ratio = 0.1) and 25 mL of NaBH_4_ (45 mmol/L) was added in the PP@AgNPs-7 sample (NaBH_4_/AgNO_3_ ratio = 11). The end-products were aqueous suspensions of nominal concentrations of 2 mmol/L in terms of Ag content, which were stored at room temperature.

The synthesis conditions are reported in Table 1.

### 2.3. Characterization Techniques

UV-vis-NIR spectra of the PP@AgNPs samples were acquired at room temperature via a Shimadzu UV-1800 (Shimadzu USA Manufacturing, Inc., Canby, OR, USA) spectrophotometer, using silica cuvettes of different pathlengths.

PP@AgNPs were investigated via field-emission scanning electron microscopy (FE-SEM) by using a ZEISS SUPRA 40 VP (Carl Zeiss NST, GmbH, Oberkochen, DEU) microscope operating at 20 keV in both direct (in-Lens mode) and back-scattered (QBSD mode) configurations. Before the analysis, the nanoparticle-containing samples were thinly sputter-coated with carbon, using a Polaron E5100 sputter coater (2M Strumenti, Rome, Italy) to obtain good electrical conductivity. Size measurements of metal clusters and silver nanoparticles were carried out by means of the open-source Image J^TM^ software on no less than 100 specimens from images at different magnifications.

The rheological characterization of the PP@AgNPs suspensions was carried out using a rotational rheometer, Physica MCR 301 (Anton Paar GmbH, Graz, Austria). The experimental temperature was set at 25.0 ± 0.2 °C by means of a Peltier heating system coupled with a solvent trap kit to prevent the solvent evaporation. A cone–plate geometry (CP50) with a diameter of 50 mm, an angle of 1° and a truncation of 99 μm was used. 

The morphological characterization of the PP@AgNP layers after deposition was carried out using AFM NTMDT NTEGRA (NT-MDT Spectrum Instruments Llc. Moscow, Russia) operating in contact and tapping mode (NSG10 with cantilever resonant frequency: 150–300 kHz). Local surface potential/work function was determined via AFM operating in Kelvin mode with conductive tips (NSG10/Pt). The electrical sheet resistance measurements of the films were performed by using a Jandel Multiheight Four Probe Head and a Keithley 2601 A System Source Meter with the appropriate acquisition software.

The electrical measurements of the films were realized at ambient conditions with a Quantum Design magnetometer. UV-Vis-NIR spectra were acquired on an Agilent Cary 60 instrument, with baseline in air.

### 2.4. Layer Deposition and Prototype Fabrication

Organic solar cells were fabricated with the following configuration: glass/Orgacon/PP@AgNPs-6/P3HT:PCBM/Field’s metal (FM), where the bilayer Orgacon/PP@AgNPs-6 works as the anode. For comparative studies, ITO slides were also used. The ITO or glass slides were cleaned in a Branson ultrasonic bath with different solvents—(1) methylene chloride (10 min), (2) isopropyl alcohol (10 min) and (3) acetone (10 min)—then dried at room temperature and exposed to UV-Ozone treatment for 10 min. All the layers were deposited via spin coating with a Laurell spin coater. For the anode preparation, Orgacon was first spun on the glass substrates, at 1000 rpm for 2 min, followed by 1 min at 4000 rpm and then annealed at 125 °C for 10 min. The following PP@AgNPs-6 layer was then deposited with the same spinning conditions but the thermal treatment at 125 °C was conducted for just 2 min. The active layer was prepared from a 1:1 wt chlorobenzene solution of P3HT:PC61BM. Both materials were dissolved in chlorobenzene with a total concentration of 20 mg/mL and kept overnight at 60 °C under magnetic stirring conditions. The mixture was then spun at 1200 rpm for 50 s with an acceleration of 2500 rpm. After deposition, the film was thermally annealed at 150 °C for 25 min. FM as the cathode was deposited at 70 °C. The I-V curves were obtained using a workstation X100 Source Measure Unit from Ossila by illuminating the devices from the ITO side with a 100 mW cm^2^ white light from a Solar Light Co. (Glenside, PA, USA) Model XPS 400 solar simulator with a Xenon lamp and AM1.5 filter. All the cells were prepared and measured under ambient conditions.

## 3. Results and Discussion

### 3.1. Characterization of Silver Nanoparticles Stabilized with PEDOT:PSS (PP@AgNP) Suspensions

At first, the samples were characterized by means of UV-Vis-NIR spectroscopy. The spectra acquired from the corresponding diluted aqueous suspensions are shown in Figure 1. In detail, Figure 1a reports the spectra of samples obtained from the first synthesis, whereas Figure 1b,c report the spectra of samples obtained from the second and the third syntheses, respectively. The spectra obtained from the diluted aqueous suspensions of the starting PEDOT:PSS formulations are reported in Appendix A for comparison.

All spectral profiles are dominated by the intense, well-defined plasmonic band at 400 nm, which confirms the formation of AgNPs within the PEDOT:PSS matrix. As far as the products from the first synthesis are concerned, the narrowest spectral lineshape of PP@AgNPs-3 (Figure 1a, green line) agrees with the increased size homogeneity of the metal nanoparticles in the sample. Conversely, the polaronic band of PEDOT, usually positioned around 900 nm, is not detectable for PP@AgNPs-2 and PP@AgNPs-3 and is only slightly hinted at for PP@AgNPs-1. These results are indicative of a partial doping loss of the conductive polymer induced by the high concentrations of the reductant NaBH_4_ [27]. Moreover, the AgNPs synthesized through “in situ” protocols are supposed to be preferentially stabilized with the PSS excess in the sample [27]. 

Further information can be derived from the spectral lineshapes of PP@AgNPs-4 and PP@AgNPs-5 in Figure 1b and of PP@AgNPs-6 and PP@AgNPs-7 in Figure 1c. The spectra of PP@AgNPs-4 and PP@AgNPs-5 are almost overlapped to highlight that PEDOT:PSS formulation negligibly affects the synthesis reaction of the AgNPs. Moreover, the lineshape of the plasmonic bands are very similar to that of PP@AgNPs-3, confirming that more homogeneous nanoparticles can be obtained by lowering the NaBH_4_/AgNO_3_ molar ratio under the same experimental conditions. In the spectral region corresponding to PEDOT absorptions, a weak shoulder at around 800 nm is observed in both cases, attributable to the polaron band of PEDOT being subjected to conformational changes [27]. This result indicates that the AgNPs are formed in the domains of the conductive polymer due to the high concentration of silver precursor in the reaction mixture, even if the lowest NaBH_4_/AgNO_3_ molar ratio is used. This hypothesis seems to be confirmed by the spectral profiles of PP@AgNPs-6 and PP@AgNPs-7, obtained by using a 20-fold lower AgNO_3_ concentration. In these cases, the polaron band is positioned at around 930 nm, which is typical of unperturbed PEDOT. It is reasonable to assume that at a low metal precursor concentration, the nanoparticles are formed in smaller quantities and outside the conductive domains, even in PEDOT:PSS formulations with low PSS content such as Clevios, and independently from the reductant concentration. However, the reaction yields drastically increase when increasing the NaBH_4_/AgNO_3_ molar ratio, as shown by the higher intensity of the plasmonic band of PP@AgNPs-7.

In addition, high-resolution FE-SEM images of PP@AgNP suspensions, acquired in both direct (left) and back-scattered (right) configurations, are shown in Figure 2 for PP@AgNPs-1, PP@AgNPs-2, PP@AgNPs-3, PP@AgNPs-6 and PP@AgNPs-7. As the comparison between the UV-Vis-NIR spectra of the samples PP@AgNPs-4 and PP@AgNPs-5 allows overlapping to sample PP@AgNPs-3, only the latter sample has been further characterized with high-resolution FE-SEM. The morphological parameters extracted from the images are listed in Table 2.

The images in Figure 2a–c highlight the presence of nearly star-shaped metal clusters of dimensions greater than or equal to 200 nm. The aggregates are composed of elongated (aspect ratio around 1.4) AgNPs arranged radially around the center of the cluster. As expected, the nanoparticles grow in average dimensions and size homogeneity from PP@AgNPs-1 to PP@AgNPs-3, that is, with the NaBH_4_/AgNO_3_ molar ratio decreasing in the synthesis reaction. Moreover, it is important to notice that the observed metal clusters in Figure 2a could represent the nanoparticle aggregates formed out of the conductive PEDOT:PSS domains. This remark is consistent with the UV-Vis-NIR spectra observed for sample PP@AgNPs-1 in Figure 1a.

Different features are observed in Figure 2d,e. Elongated nanoparticles of reduced average size (about 20 nm) and axial ratio (1.3) are highlighted, with better dispersed and organized in clusters of dimensions around 70 nm only in the case of PP@AgNPs-6. This result is ascribable to the 20-fold lower quantity of the silver salt precursor used in the synthesis of this sample.

Finally, the rheological behavior of the samples was also investigated to achieve information about the film-forming capability of the AgNP-containing suspensions [28]. As an example, the curves of viscosity (η) obtained from PP@AgNPs-1, PP@AgNPs-2, PP@AgNPs-3, PP@AgNPs-6 and PP@AgNPs-7 are reported in Figure 3. The main rheological parameters extracted from the curves are listed in Table 3.

The three samples show the characteristic viscosity curves of pseudoplastic Bingham fluids, which are as expected in that the suspensions contain aggregated, non-deformable nanoparticles. The absolute viscosity values at shear rate equal to zero (η_0_), evaluated via extrapolation from the regression data model supplied by the instrument, are generally low. In detail, they increase from PP@AgNPs-1 to PP@AgNPs-3. These results are consistent with the differences in the AgNP size found in the samples, since larger, non-deformable nanofillers are expected to further increase the η_0_ value of the fluid in which they are dispersed and extend its Newtonian behavior. The trends with the shear rate of Figure 3b, corresponding to PP@AgNPs-6 and PP@AgNPs-7, are almost superimposed and settle for lower η values. These results were expected given that the samples are characterized by a reduced concentration of particles with comparable size. Taking into account this study, the PP@AgNP_6 and _7 are the most promising samples for making films.

### 3.2. Deposition of the PP@AgNP Suspensions 

The first deposited layer onto glass substrates is constituted by an Orgacon layer in place of ITO, and the AFM images of this layer are shown in Figure 4a (root mean square, RMS: 1.2 nm; Table 4). The choice to use Orgacon onto glass is due not only to electronic reasons in order to guarantee high conductance in the absence of ITO, but also to improve the adhesion of the following PP@AgNP layer. At the same time, it also provides a flat and homogeneous layer for the subsequent active layer deposition, as shown in Figure 4f. 

#### 3.2.1. Effect of Reducing Agent/AgNO_3_ Ratio

In the PP@AgNPs-1 sample displaying a higher NaBH_4_/AgNO_3_ molar ratio, elongated structures with lengths ranging from 400 nm up to 1 µm can be observed (Figure 4b), over a quite smooth layer. The presence of crystallite-like structures is responsible for the large corresponding RMS of 33 nm (Table 4). When considering the FE-SEM analysis (Figure 2a) and based on the proposed mechanism for stabilizing the nanostructures, it is likely that the elongated structures correspond to the silver nanoparticles aggregates directly covered with the excess of PSS and embedded in the PEDOT matrix that forms a smooth layer, similarly to the underlying film. Otherwise, the morphologies of PP@AgNPs-2 (Figure 4c) and PP@AgNPs-3 (Figure 4d) are granular-type, with a distribution of spherical grains with a mean diameter of 63 nm (PP@AgNPs-2) and 84 nm (PP@AgNPs-3), respectively. Moreover, the roughness of the PP@AgNP layer decreases from 33 nm for PP@AgNPs-1 to 4.3 nm and 5.3 nm for PP@AgNPs-2 and PP@AgNPs-3, respectively (Table 4). It is to be noted that a similar morphology was observed for PP@AgNPs-5 (see further) prepared with a NaBH_4_/AgNO_3_ molar ratio that is even lower, resulting in a lower RMS of 4.4 nm. This trend could indicate that the AgNPs are better dispersed in the PEDOT:PSS matrix along with the decrease in the NaBH_4_ content during the sample preparation. 

#### 3.2.2. Effect of the Formulation of PEDOT:PSS Suspension

In the second series of samples, a comparison between the two typologies of commercial PEDOT:PSS within the nanocomposite blend (Clevios and Orgacon for PP@AgNPs-4 and PP@AgNPs-5, respectively) is carried out, while the NaBH_4_/AgNO_3_ molar ratio is kept constant (Table 1). Figure 4e,f report the AFM images of the PP@AgNPs-4 and PP@AgNPs-5 samples, respectively. It is possible to observe the formation of aggregates over a rough texture, similarly to PP@AgNPs-3, giving rise to an RMS of 7.7 nm for the Clevios-based sample and 4.4 nm using the Orgacon one (Figure 4e,f, respectively; Table 4). As already observed from the absorption spectra, the morphology study confirms that the formulation of the PEDOT:PSS aqueous suspension does not significantly affect the synthesis of AgNPs.

Once again in agreement with the spectroscopic characterization, the images of Figure 4d–f indicate that AgNPs were also formed in the domains of the conductive polymer PEDOT, notwithstanding the lowest NaBH_4_/AgNO_3_ molar ratio adopted in the cases of PP@AgNPs-4 and PP@AgNPs-5, most likely because of the high concentration of silver precursor in the reaction medium.

#### 3.2.3. Effect of the AgNO_3_ and PEDOT:PSS Concentration in the Reaction Medium

In the third sample series, PP@AgNPs-6 and 7, the influence of the medium concentration was tested. The ratio of Ag/PEDOT:PSS was almost doubled with respect to the previous ratio (0.1 mmol/mL PEDOT vs. 0.06 mmol/mL). For the PP@AgNPs-7 sample with a NaBH_4_/AgNO_3_ molar ratio of 11, the AgNP layer is highly rough compared to the other samples with an RMS of 8.1 nm (Table 4) and is characterized by big clusters with a diameter reaching 1 µm (Figure 4g). This is probably due to the coalescence of small AgNPs and PEDOT:PSS aggregates. On the other hand, in the case of the PP@AgNPs-6 sample prepared with a NaBH_4_/AgNO_3_ molar ratio equal to 0.1, the lack of the reducing agent (BH_4_^-^) with respect to the silver salt gives rise to a flat network with interconnected nanoparticles, responsible for the lowest roughness (Figure 4h and RMS = 1.3 nm Table 4). Hence, an excess of silver salt in the starting reagents induces a coalescence of reduced AgNPs instead of a homogenous distribution of smallest AgNPs.

From these remarks, it is safe to infer that the combination of a low NaBH_4_/AgNO_3_ ratio and a lower concentration of silver ions and PEDOT:PSS in the reaction medium leads to the production of the most homogeneous AgNP layer, characterized by the highest AgNP interconnection and flatness (as for PP@AgNPs-6), while an excess of the reducing agent NaBH_4_ and/or a high concentration of silver ions produces big aggregates and a rougher layer (as for PP@AgNPs-1).

### 3.3. Solar Cells Prototypes: Preliminary Study

As a proof of concept to demonstrate the potentialities of PP@AgNPs, a preliminary polymer solar cell was fabricated using the PEDOT:PSS/AgNP layer as the anode, and its performance was compared with devices with the same active layer and the classic ITO anode. As discussed previously, the first PEDOT:PSS layer is required for a continuous and smooth deposit of the next nanoparticle-based layer. Among all the nanoparticles synthesized, the PP@AgNP-6 sample was chosen because of (i) its high morphological quality in that the RMS roughness is minimized compared to all the other cases; (ii) the UV-Vis-NIR features in which the polaron band of PEDOT:PSS is still observable, revealing the absence of doping losses or conformational changes that could affect the electrical conductivity; (iii) in the PP@AgNP-6 sample, the AgNPs are expected to be formed outside the conductive domains of PEDOT and then to act as electrical bridges between one domain and another, as mentioned in previous works [26,27]; (iv) from preliminary electrical sheet resistance measurements with the four-probe method, PP@AgNPs-6 results to have a sheet resistance (ohm/cm^2^) sensibly lower than the other samples (i.e., <10^3^ ohm/cm^2^ for PP@AgNPs-6 against values above 10^4^ ohm/cm^2^ for all the other samples), probably due to the higher morphological homogeneity and flatness of PP@AgNPs-6 compared to the other samples.

The anode glass/Orgacon/PP@AgNPs-6 presented a σ value of 1.61 × 10^−2^ S/cm, an order of magnitude greater than the neat Orgacon layer, suggesting that the incorporation of the AgNPs allows for better electrical continuity among the conductive PEDOT chains. 

Before preparing the device, the transmittance of the anode was evaluated. Figure 5 shows the UV-Vis-NIR spectrum of the Orgacon/PP@AgNP-6 film and, for the sake of comparison, those of the Orgacon layer, the glass substrate and of ITO slides with two different resistances (8–12 Ωsq and 30–60 Ωsq).

The glass substrate presents a quite constant and strong transmittance (87–90%) from 1100 nm up to 370 nm. At shorter wavelengths, the transmittance starts to decrease and finally falls to zero at 260 nm. The commercial ITO slides present some absorptions in the visible region. In particular, the slides with 30–60 Ωsq resistance exhibit interference fringes, likely suggesting a quite thin thickness. In general, ITO transmittance is a bit lower (≈75–80%) than that of glass, in agreement with the presence of a tin oxide layer. The transmittance of the PP@AgNP-6 anode has a value of 74–80% between 300 and 500 nm and then decreases to ≈60% because of the PEDOT polaron absorption, showing behavior similar to that of the corresponding Orgacon layer. At 446 nm, a weak valley that corresponds to the plasmon silver absorption can be observed, as better visualized in the absorption spectrum of the figure inset. The red shift from the value of 400 nm found for the corresponding aqueous suspension of Figure 2c is consistent with the cluster formation in the solid state, as previously observed for other AgNP–PEDOT composites [20]. Despite the slight decrease, the T% value in the visible range is reasonably satisfactory for an anode.

In the anode characterization, work function (WF) measurements have an important role, since WF position determines the energy needed for the hole extraction process from the active layer to the anode itself. Usually, when ITO is used as an anode, a PEDOT layer is deposited in order to increase the WF from 4.80–5.00 to 5.10 eV to better match the HOMO level of poly(3-hexylthiophene) (P3HT) in the active layer (5.1 eV) and thus favor the hole extraction from donor to anode. In our case, ITO is substituted by the layer of Orgacon/PP@AgNP-6. Since Orgacon has a WF 4.87 eV, an increase in the WF is still needed, and AgNPs partially satisfy this requirement, raising the level to 4.95 eV.

Considering all these properties, we fabricated the device by using the 1:1 blend of P3HT with PC_60_BM as the active layer, which is a performing OPV system commonly used in our laboratory as a benchmark. 

Figure 6 shows the J-V curves for the solar cell, the device configuration of which is represented in the inset. Reference cells were also prepared by using ITO with different resistances (Appendix A). The overall photovoltaic parameters are displayed in Table 5.

It can be observed that the open circuit V_OC_ of all the devices is almost constant, corresponding to 0.5–0.6 V. Since this value is related to the difference between the HOMO level of the electron donor (P3HT) and the LUMO level of the acceptor (PC61BM), it is expected to have the same value in all the devices. Conversely, the short circuit current J_SC_ and the fill factor FF differ significantly, exhibiting a much lower value for the cell with the nanostructured anode. As FF is usually associated with the morphological quality of the active layers, the observed decrease can be explained by the fact that the active layer will be deposited on the subjacent substrate following its roughness. 

On the other hand, the lower J_SC_ can be attributed to the larger resistivity of the anode. In fact, the ITO substrates, with the nominal resistance of 8–12 and 30–60 Ωsq according to the provider, give a ρ value of 0.20 Ωcm and 1.09 Ωcm when measured via magnetometry, whereas the nanostructured anode under study shows a ρ value of 62 Ωcm under the same conditions. Moreover, the fact that the performance of the cells deteriorates as the resistance of the two ITO-based reference solar cells increases seems to indicate that the electrical property of the anode is the most important parameter yet to be optimized for the composite based on AgNPs and PEDOT. It is worth mentioning that, in this preliminary assay, devices were fabricated and analyzed in ambient conditions. Moreover, a room-temperature processable cathode (Field’s metal) was used to make the process cheaper, easier and environmentally competitive.

## 4. Conclusions

Silver nanoparticles embedded in PEDOT:PSS (PP@AgNPs) were synthesized via the chemical reduction of silver nitrate with NaBH_4_, in the presence of commercial PEDOT:PSS, varying the following reaction conditions: NaBH_4_/AgNO_3_ molar ratio, PEDOT-PSS formulation (Orgacon or Clevios) and the concentration of both AgNO_3_ and PEDOT:PSS in the medium. The best-dispersed nanoparticles corresponded to the sample obtained with the lowest silver content, lowest NaBH_4_/AgNO_3_ molar ratio and highest dilution of the PEDOT:PSS in the formulation and concentration, labeled as PP@AgNP_6.

The combined investigation carried out using UV-Vis-NIR spectroscopy and the FE-SEM technique revealed that the best-dispersed nanoparticles corresponded to the sample obtained with the lowest silver content, lowest NaBH_4_/AgNO_3_ molar ratio and highest dilution of the PEDOT:PSS in the formulation and concentration, labeled as PP@AgNP_6. In this case, AgNPs are rather formed outside the PEDOT complex domains and stabilized by the PSS in excess, which in turn guarantees higher electrical conductivity since the conjugation in the conducting polymer turns out to not be affected by the presence of the nanoparticles.

Accordingly, the rheological study of the sample hydrosol showed pseudoplastic Bingham behavior with the lowest viscosity value, which was expected to correspond to the best filmability properties. The AFM images confirmed that the best homogeneity in terms of flatness and nanoparticles interconnection was obtained with the PP@AgNP_6 specimen.

Based on all the results, the PP@AgNP_6 sample was selected to prepare ITO-free anodes for preliminary photovoltaic studies with P3HT:PCBM as the active layer. We remark that we used Field metal as the cathode, which is a low-temperature melting material, to give a further ecological benefit to the fabrication process. Under our conditions, and comparative to ITO-based reference cells, the open circuit voltage (V_OC_) was maintained but the fill factor (FF) and short circuit current (J_SC_) were lower, because of the higher resistivity of the nanostructured anode and morphological poorer quality of the active layer deposited on it. However, the preliminary value of 0.03% obtained for the efficiency confirms the potentiality of a nanostructured, water-processable anode in the fabrication of new-generation solar cells after the proper optimization of both AgNP morphology and device configuration.

## Figures and Tables

**Figure 1 polymers-15-03675-f001:**
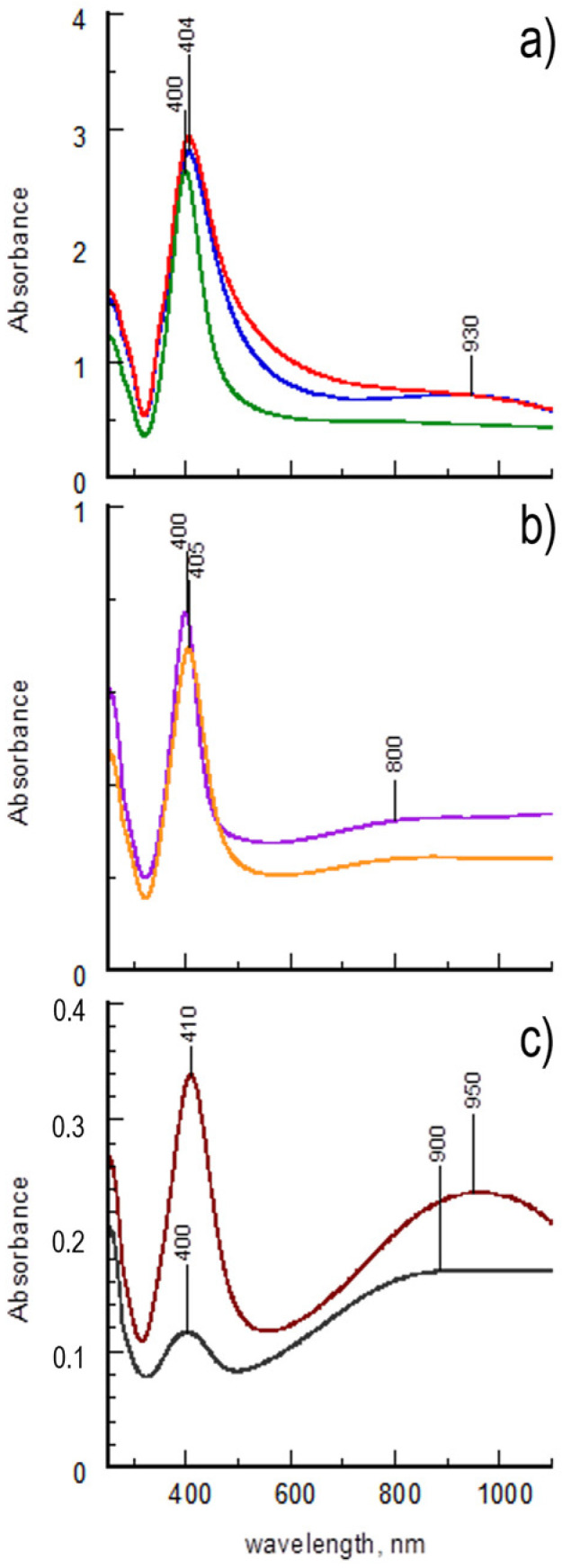
UV-Vis-NIR spectra of diluted suspensions. From top to bottom: (**a**) PP@AgNPs-1 (blue line), PP@AgNPs-2 (red line) and PP@AgNPs-3 (green line); (**b**) PP@AgNPs-4 (purple line) and PP@AgNPs-5 (orange line); (**c**) PP@AgNPs-6 (gray line) and PP@AgNPs-7 (brown line).

**Figure 2 polymers-15-03675-f002:**
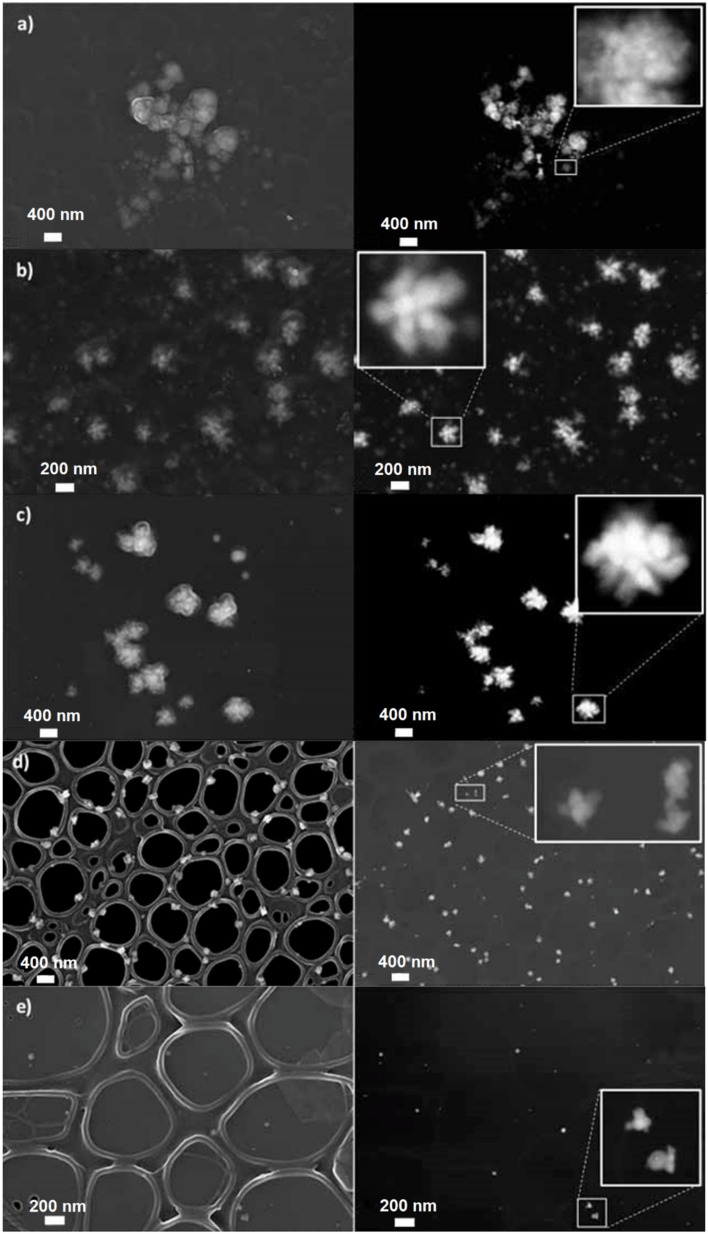
FE-SEM images acquired in direct (In-Lens, left) and back-scattered (QBSD, right) configurations from the samples of PP@AgNPs, for all the images (**a**–**e**). From top to bottom: (**a**) PP@AgNPs-1, (**b**) PP@AgNPs-2, (**c**) PP@AgNPs-3, (**d**) PP@AgNPs-6 and (**e**) PP@AgNPs-7. Magnifications of structural details are reported in the insets of the QBSD pictures.

**Figure 3 polymers-15-03675-f003:**
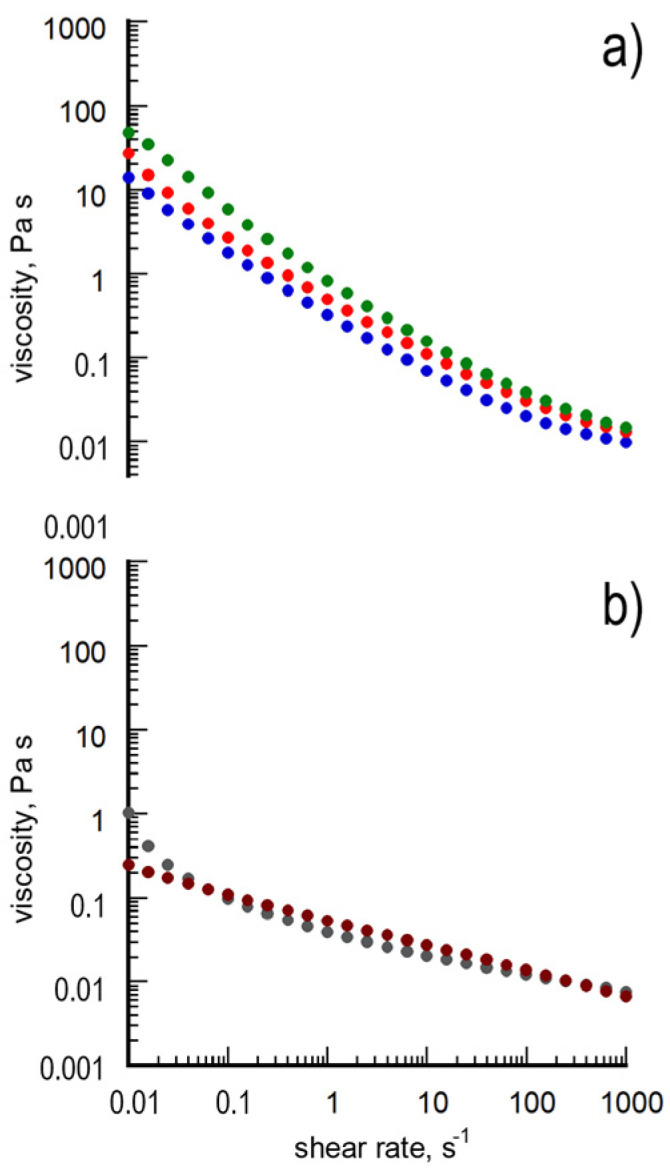
Curves of viscosity obtained from samples. From top to bottom: (**a**) PP@AgNPs-1 (blue full circles), PP@AgNPs-2 (red full circles) and PP@AgNPs-3 (green full circles); (**b**) PP@AgNPs-6 (gray full circles) and PP@AgNPs-7 (brown full circles).

**Figure 4 polymers-15-03675-f004:**
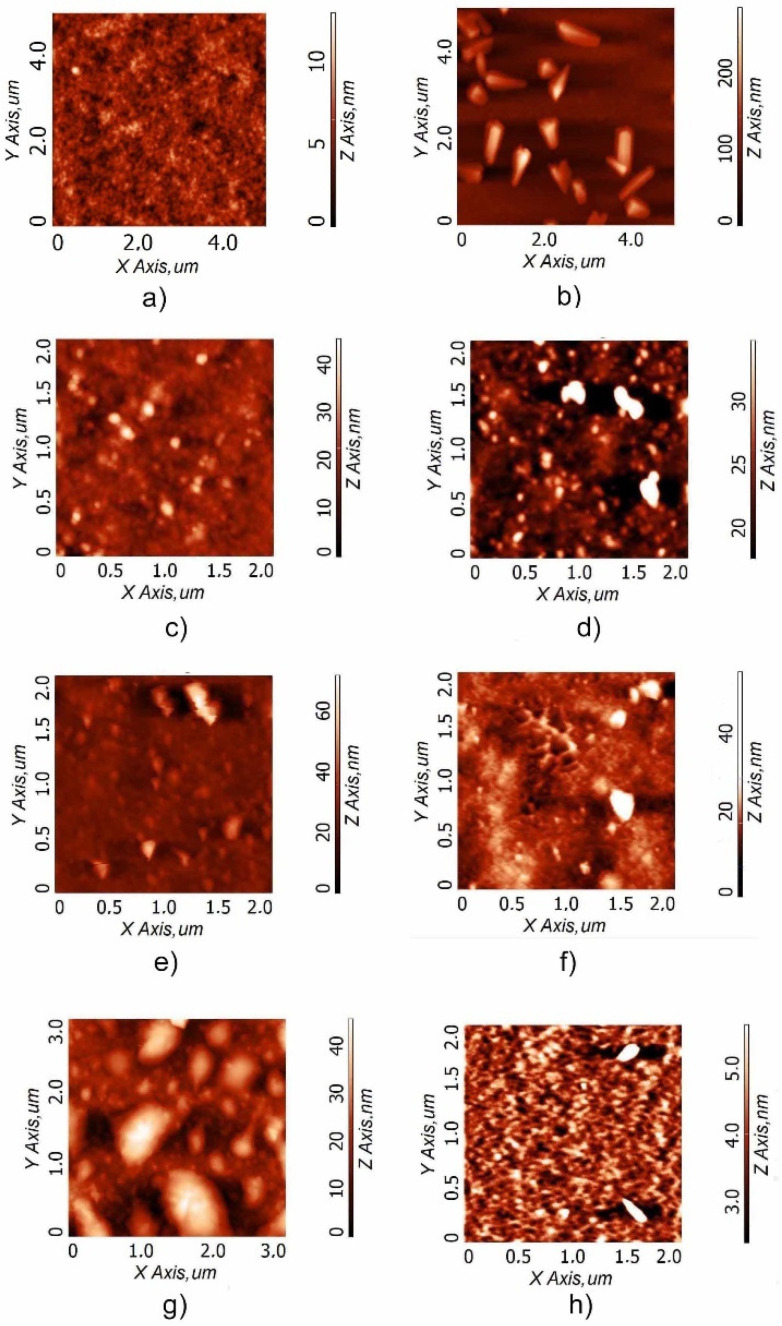
Atomic force microscopy (AFM) images of the PP@NP samples: (**a**) Orgacon (RMS = 1.2 nm); (**b**) PP@AgNPs-1 (RMS = 33 nm); (**c**) PP@AgNPs-2 (RMS = 4.3 nm); (**d**) PP@AgNPs-3 (RMS = 5.3 nm); (**e**) PP@AgNPs-4 (RMS = 7.7 nm); (**f**) PP@AgNPs-5 (RMS = 4.4 nm); (**g**) PP@AgNPs-07 (RMS = 8.1 nm); (**h**) PP@AgNPs-6 (RMS = 1.3 nm).

**Figure 5 polymers-15-03675-f005:**
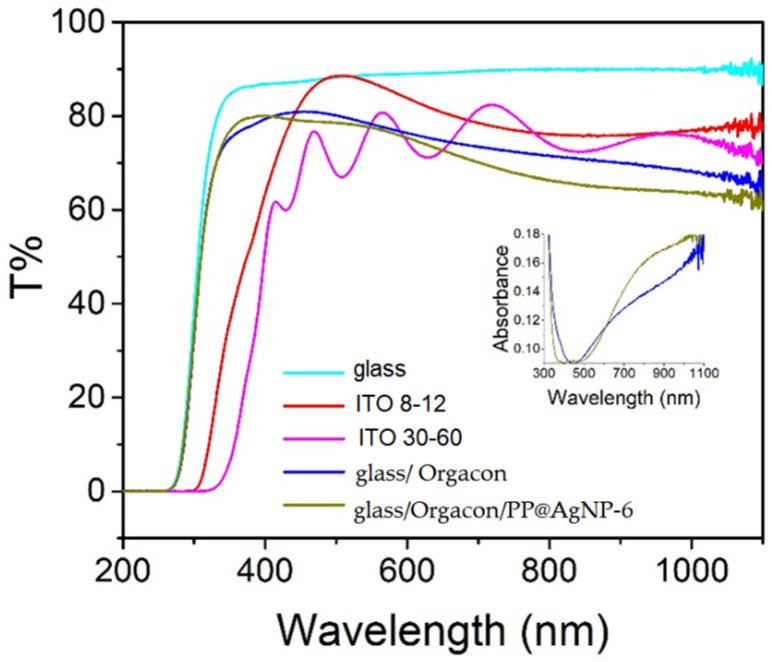
UV-Vis-NIR spectra in transmittance mode of glass/Orgacon/PP@AgNP-6 and, for comparison, the glass/Orgacon layer, the glass substrate and two commercial ITO slides with different resistances. Inset: corresponding absorbance spectra of the glass/Orgacon/PP@AgNP-6 and Orgacon layers on glass.

**Figure 6 polymers-15-03675-f006:**
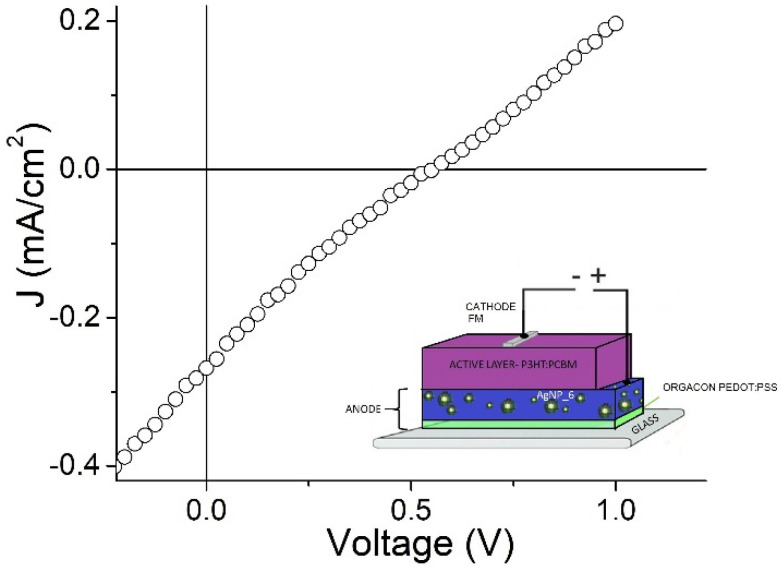
J-V curve for the solar cell under study, the configuration of which is sketched in the inset.

**Table 1 polymers-15-03675-t001:** Experimental conditions set up for the synthesis of PP@AgNPs.

Sample	PEDOT:PSS Formulation	NaBH_4_/AgNO_3_ Molar Ratio	Ag Conc (mmol/L)
PP@AgNPs-1	Orgacon	9.9	40
PP@AgNPs-2		4.9	40
PP@AgNPs-3		1.0	40
PP@AgNPs-4		0.1	40
PP@AgNPs-5	Clevios	0.1	40
PP@AgNPs-6		0.1	2
PP@AgNPs-7		11.0	2

**Table 2 polymers-15-03675-t002:** Summary of the PP@AgNP morphological parameters measured from FE-SEM images of Figure 2.

Sample	Cluster Range Size (nm)	Cluster Average Size (nm)	Nanoparticle Average Size (Standard Deviation) (nm)	Nanoparticle Aspect Ratio (Standard Deviation)
PP@AgNPs-1	90–500	-	26 ± 1 (7)	1.44 ± 0.06 (0.41)
PP@AgNPs-2	120–300	220 ± 10	32 ± 1 (9)	1.39 ± 0.04 (0.31)
PP@AgNPs-3	60–600	200 ± 5	60 ± 1 (15)	1.45 ± 0.04 (0.45)
PP@AgNPs-6	30–160	240 ± 20	21 ± 1 (5)	1.31 ± 0.04 (0.25)
PP@AgNPs-7	-	70 ± 1	18 ± 1 (9)	1.26 ± 0.03 (0.24)

**Table 3 polymers-15-03675-t003:** Summary of the rheological parameters extracted from the curves of viscosity in Figure 3.

Sample	η_0_ (Pa s)
PP@AgNPs-1	14.1
PP@AgNPs-2	26.7
PP@AgNPs-3	47.2
PP@AgNPs-6	1.0
PP@AgNPs-7	0.2

**Table 4 polymers-15-03675-t004:** Summary of the root mean square (RMS) related to the synthesis conditions of the samples extracted from Figure 4.

Sample Label	NaBH_4_/AgNO_3_ Molar Ratio	Ag Conc (mmol/L)	Root Mean Square, RMS (nm)
PEDOT:PSS (Orgacon)	-	-	1.2
PP@AgNPs-1	9.9	40	33.0
PP@AgNPs-2	4.9	40	4.3
PP@AgNPs-3	1.0	40	5.3
PP@AgNPs-4	0.1	40	7.7
PP@AgNPs-5	0.1	40	4.4
PP@AgNPs-6	0.1	2	1.3
PP@AgNPs-7	11.0	2	8.1

**Table 5 polymers-15-03675-t005:** Photovoltaic properties of organic solar cells studied in this work.

Electrode	V_oc_ (Volts)	J_sc_ (mA/cm^2^)	FF	PCE (%)
ITO 8–12 Ωsq	0.60	5.13	0.55	1.70
ITO 30–60 Ωsq	0.53	3.16	0.30	0.51
Orgacon/PP@AgNP-6	0.55	0.27	0.21	0.03

## Data Availability

Not applicable.

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
