# Peer review of "Silver Nanoparticle–PEDOT:PSS Composites as Water-Processable Anodes: Correlation between the Synthetic Parameters and the Optical/Morphological Properties"

_polymers, 2023, doi:10.3390/polym15183675_

Round 1

Reviewer 1 Report

- DLS would give a better assessment of the size of the prepared nanoparticles, as the number of the measured NPs/clusters by SEM is limited. Zeta potential measurement for the prepared NPs and their clusters will be a good addition to the characterization, and it can be studied under different conditions to prove or test some of their properties. 

- It might be worth investigating the effect of plasticizers on the film formability of the NPs suspensions

- The conclusion is a bit long and more of a summary, it includes unnecessary details, and it must show the significance of the work and what it adds to the accumulating literature.  

- Simple grammatical mistakes better be corrected such as

Line 147: rewrite as follows: 0.1 g of AgNO3 were added.

Line 154: simple past tense fits better than the present perfect tense. 

Author Response

Please see the point-by-point response to Reviewer #1's comments in the attachment.

Reviewer 2 Report

In the manuscript, the Authors present the optimization strategy for improvement of the performance of PEDOT:PSS-AgNPs composite materials, for the ultimate goal which is the design of a prototype of a solar cell. The manuscript is well written, but some parts needs to be improved before accepting for publication:

1. The abstract should not contain abbreviations (PC60BM).

2. Why there are no data for samples 4 and 5 in Fig1 and Fig 3?

3. Fig.1: what are the "circles" visible in d and e?

4. Fig.1a-c: I would prefer not to use the name "spherical" for the shapes in these images - the aggregates look more like stars.

5. Fig.2: UV-vis spectra of both forms of pristine PEDOT:PSS should be present for comparison.

6. In few places, the Authors discuss the conductivity of samples - therefore, it would be recommended to perform an extensive analysis of conductivity of produced materials, e.g. with the use of EIS.

7. Fig.5: Figure legend should be simplified- instead of product code it would be better to write "Orgacon".

 Minor editing of English language required.

Author Response

(The authors gave the same response as above.)

Round 2

Reviewer 2 Report

The revised version of the manuscript is suitable for publication.